# PedSleepMAE: Generative Model for Multimodal Pediatric Sleep Signals

1st Saurav Raj Pandey
*Department of Computer Science*
*UNC Chapel Hill*
Chapel Hill, NC, USA
srpandey@unc.edu

2nd Aaqib Saeed
*Department of Industrial Design*
*Eindhoven University of Technology*
Eindhoven, Netherlands
a.saeed@tue.nl

3rd Harlin Lee
*School of Data Science and Society*
*UNC Chapel Hill*
Chapel Hill, NC, USA
harlin@unc.edu

*Abstract*—**Pediatric sleep is an important but often overlooked area in health informatics. We present PedSleepMAE, a generative model that fully leverages multimodal pediatric sleep signals including multichannel EEGs, respiratory signals, EOGs and EMG. This masked autoencoder-based model performs comparably to supervised learning models in sleep scoring and in the detection of apnea, hypopnea, EEG arousal and oxygen desaturation. Its embeddings are also shown to capture subtle differences in sleep signals coming from a rare genetic disorder. Furthermore, PedSleepMAE generates realistic signals that can be used for sleep segment retrieval, outlier detection, and missing channel imputation. This is the first general-purpose generative model trained on multiple types of pediatric sleep signals.**

*Index Terms*—**sleep, pediatric health, polysomnography, eeg, apnea, generative AI, masked autoencoder**

## I. Introduction

Sleep is vital to the health and well-being of infants, children, and adolescents. Sleep disorders and disturbances can affect childhood neurobehavioral development and cognitive functions and even cause morbidity in severe cases of obstructive sleep apnea (OSA) [1]. However, pediatric sleep has been often overlooked in health informatics, partially due to underreporting, underdiagnosis [1] and lack of public datasets. Most importantly, there is a common misconception that pediatric sleep is not distinct from adult sleep, when in fact children have unique physiology and therefore must be studied separately from adults, including for computational models. For example, [2] found that popular sleep stage classification models based on adult data struggle to generalize well on patients younger than 10 years old.

The clinical gold standard for sleep medicine is currently based on polysomnography (PSG) or an overnight sleep study. Since sleep is a complex process that involves multiple organ systems, PSG data encompass many channels and modalities, such as electroencephalography (EEG), respiratory signals and sometimes videos. During PSG, a sleep technician monitors the signals and manually annotates events such as movement, coughing, breathing changes and sleep stages (e.g. REM). Compared to adult counterparts, software for automated detection of such sleep events is not as widespread (or verified and trusted) in pediatric sleep clinics, making this a labor-intensive process. Combined with the rich PSG data, this strongly motivates the use of machine learning in pediatric sleep research.

In the past few years, deep learning-based generative models and self-supervised learning (SSL) have come to the forefront of machine learning. Unlike supervised learning, SSL trains the model to do auxiliary tasks on unlabeled data (e.g. predict whether an image is rotated). Once informative embeddings or features are learned, SSL only then uses labeled data for downstream classification or regression tasks of interest. This approach can fully leverage large unlabeled datasets and have been shown to generalize well to unseen data, beating supervised models in some cases [3]. Considering the cost of acquiring gold standard labels in medicine, SSL is a particularly promising approach in health informatics [4].

This work presents PedSleepMAE (Pediatric Sleep Masked Autoencoder), a generative model for multichannel pediatric PSGs. To our best knowledge, PedSleepMAE (Fig. 1) is the first to satisfy all of the following characteristics:

- Explicitly focuses on pediatric sleep and leverages large public data collected in real clinical setting.
- Handles multiple types and channels of PSG signals, e.g. beyond EEGs [5], [6] and can be flexible with missing channels.
- Trains a transformer-based model via SSL such that the model can be used for multiple tasks, e.g. beyond sleep scoring [5] or apnea detection [7].

Our experiments are carefully designed to demonstrate interesting potential use cases of the PedSleepMAE framework:

- Automated sleep event detection and sleep scoring.
- Generate hypotheses for sleep biomarkers in conjunction with the electronic health record (EHR).
- Generate or retrieve representative examples per patient. Detect and flag outliers for clinicians or researchers.
- Impute missing channels.

In the rest of the paper, we will describe the dataset and deep learning model in Sec. II, evaluate diagnostic quality of model embeddings in Sec. III, evaluate accuracy of generated signals in Sec. IV and conclude in Sec. V.

## II. Methodology

We describe the dataset and PedSleepMAE. PedSleepMAE is open source and the full code is available at

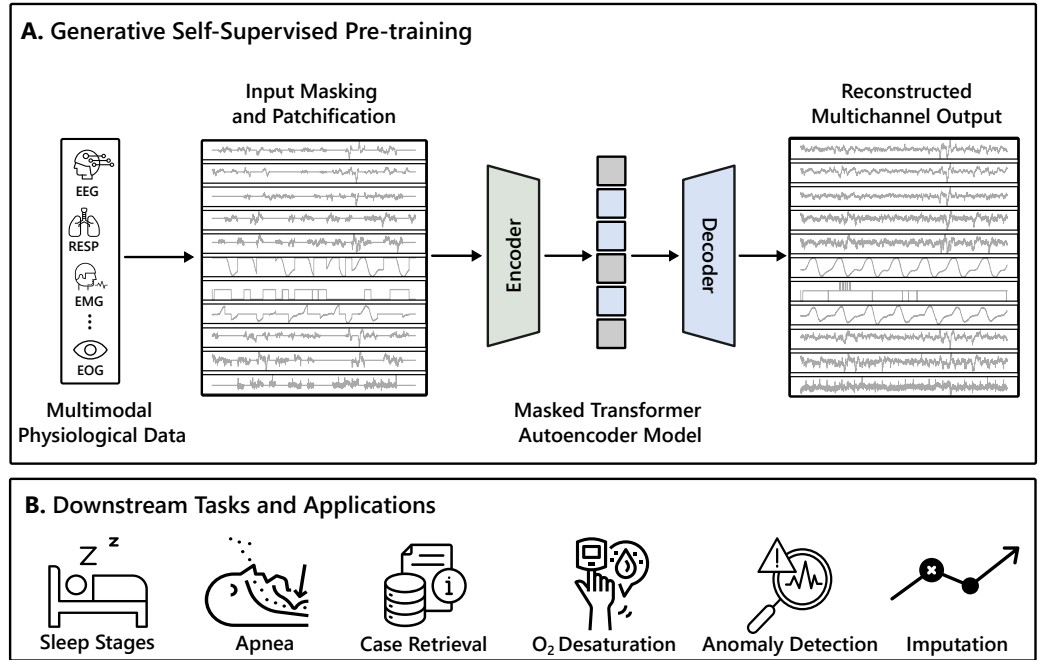

Fig. 1: An overview of PedSleepMAE framework.

### A. Polysomnography (PSG) Data

We utilize the Nationwide Children's Hospital (NCH) Sleep DataBank [8], which is a large, public, and fairly recent pediatric PSG dataset collected in a real clinical setting. Within this dataset, we analyzed 2,379 PSGs that had 16 of the most common channels [8, Table 2]. This includes 7 channels of EEG (C3-M2, O1-M2, O2-M1, CZ-O1, C4-M1, F4-M1, F3-M2), $CO_2$ level (CAPNO), $O_2$ level (SpO2), breathing effort (RESP Thoracic, RESP Abdominal), snoring (SNORE), CPAP airflow (C-FLOW), electrooculogram (EOG; LOC-M2, ROC-M1) and electromyogram (EMG; CHIN1-CHIN2). These PSGs are also accompanied by patients' electronic health records (EHR) and annotations such as sleep stages (Wake, Non-REM 1, Non-REM 2, Non-REM 3, REM), EEG arousal, apnea, and hypopnea for every 30 seconds of sleep. PSG is considered in units of 30 seconds, recorded (or downsampled) at 128Hz across 16 channels. We normalize all channels to 0 mean and 1 standard deviation for stable model training.

### B. Multichannel Masked Autoencoder (MAE)

Masked autoencoder (MAE) [9] is a state-of-the-art deep-learning-based generative model. It is based on the transformer neural network architecture [10] and has a encoder-decoder structure. During SSL, input data is divided into small patches and then randomly masked, and the MAE is trained to reconstruct the masked patches. This is similar to masked language modeling in natural language processing, e.g. BERT [11]. During this reconstruction process, the encoder learns informative non-linear features or embeddings, and the decoder learns how to generate data back from the

embeddings. MAEs are distinct from traditional autoencoders, which focus on data compression, and are much easier to train than variational autoencoders (VAEs) or generative adversarial networks (GANs) since there are no regularization terms.

Although MAE is originally proposed for images and videos, we easily adapted it to a multichannel signal setting for PSGs. Both encoder and decoder module consist of three layers of Vision Transformer [12] attention blocks with four attention heads. The attention mechanism allows the model to capture both intra- and inter-channel relationships in PSG. We used a masking ratio of 50% for SSL, and the dimension of the embeddings used in our experiments is 7,680. Appendix A describes the model and pretraining strategy in greater detail.

### III. HOW MUCH DIAGNOSTIC INFORMATION DO THE EMBEDDINGS CONTAIN?

We evaluate how much diagnostic information is in the 7680-dimensional embeddings from the PedSleepMAE encoder. The PSGs in NCH Sleep DataBank are accompanied by rich EHR and annotated with clinician-verified sleep events. We fully utilize these clinical labels to qualitatively and quantitatively measure how well these groups are separated in the embedding space.

### A. Visualization

We first employ Uniform Manifold Approximation and Projection (UMAP) [13] to reduce the embeddings into 2 dimensions and visualize them in Fig. 2. Each plot corresponds to one PSG, and each point represents 30 seconds of sleep, colored by different sleep events. PSGs are selected as follows: one PSG chosen randomly from all patients (Fig. 2a), one PSG from the top 5 with the highest apnea occurrences (Fig.

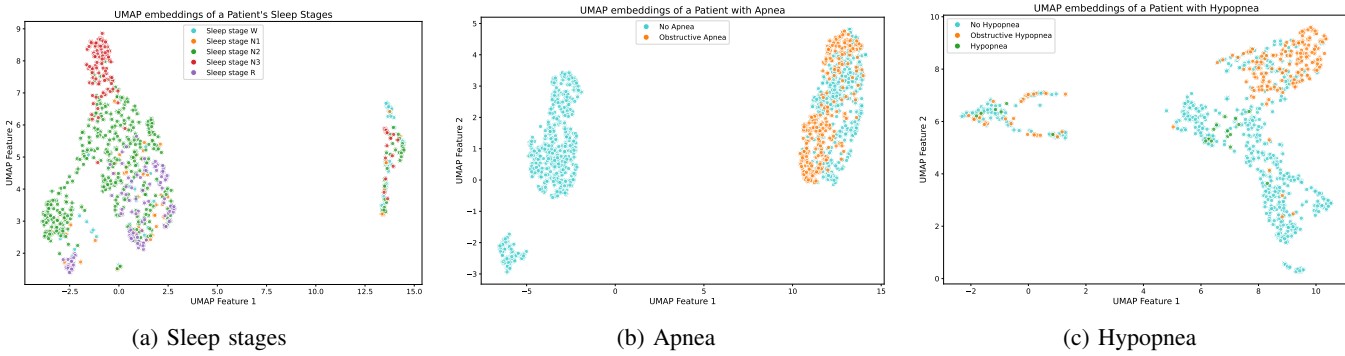

| (a) Sleep stages | (b) Apnea | (c) Hypopnea |

Fig. 2: UMAP visualizations suggest that PedSleepMAE embeddings are clustered by sleep events. Each point represents 30 seconds of sleep, and each plot corresponds to one PSG. PSGs are selected as follows: (a) one PSG chosen randomly from all patients, (b) one PSG from the top 5 with the highest apnea occurrences, and (c) one PSG from those with 5 to 30 cases of hypopnea. This was to avoid cherry-picking but still ensure there are enough sleep events and show some breadth. See more visualizations in Appendix B.

2b), and one PSG from those with 5 to 30 cases of hypopnea (Fig. 2c). This was to avoid cherry picking while still ensuring that there are enough sleep events to be plotted and also showing some breadth in our results. For readability, we defer UMAP plots of samples colored by oxygen desaturation and EEG arousal to Figs. 10 and 11 in Appendix B. Figs. 2, 10 and 11 reveal clusters in most cases. We emphasize that PedSleepMAE was able to capture characteristics of various sleep events and distill it into the embeddings despite not having seen any of these labels during pretraining.

We also applied UMAP to visualize randomly selected embeddings from multiple PSGs (Fig. 12 in Appendix B). Unlike with earlier per-PSG plots, this process did not yield distinct clusters as anticipated. We hypothesize that due to high variability and confounding variables among patients, such as differences in age, sex and underlying health conditions, clusters are not distinct enough to be seen in 2-dimensional UMAP space.

### B. Sleep Event Classification

To more quantitatively measure the separation of embeddings per sleep event (which might not be visible in 2-dimensional UMAP space), we designed 6 downstream classification tasks. They are 5-stage sleep scoring and 5 binary detection problems: oxygen desaturation, EEG arousal, apnea, hypopnea, and a combined case of apnea and hypopnea. We froze PedSleepMAE, then trained linear classifiers on top of the embeddings. The classifiers are intentionally kept as simple as possible so that we can measure the diagnostic information in the embeddings and not the classifier. See Appendix C for more details on experiment setup.

Table I and Fig. 3 present the overall accuracy, F-1 score, AUC score and confusion matrix on the test sets. For sleep scoring, we report the weighted F-1 score and weighted AUC using a one-vs-rest (OvR) approach. Our F-1 scores for all binary classification tasks are consistently higher than the corresponding percentage of positive cases, which represents

TABLE I: Test set accuracy, F-1 score and AUROC from linear probing suggest that PedSleepMAE embeddings contain information relevant to sleep health. Last column shows the percentage of positive samples for binary tasks, which is the baseline performance for a random classifier.

| Classification task | Accuracy (%) | F-1 (%) | AUC | Class 1 (%) |
|---|---|---|---|---|
| 5-stage sleep scoring | 69.2 | 71.3 | 0.899 | - |
| Oxygen desaturation | 67.9 | 28.4 | 0.773 | 8.78 |
| EEG arousal | 86.0 | 28.7 | 0.817 | 4.7 |
| Apnea | 97.6 | 10.4 | 0.797 | 0.83 |
| Hypopnea | 93.7 | 16.5 | 0.798 | 1.97 |
| Apnea-Hypopnea | 93.6 | 20.4 | 0.797 | 2.80 |

the random baseline F-1 score. Considering that there was no fine-tuning of PedSleepMAE, and the classifiers are simply fitting a hyperplane in the embedding space, the classification is successful in all cases. This is comparable to 76% sleep scoring accuracy reported by [5] and 66.8% oxygen desaturation accuracy from [6], even though they used supervised learning. However, it should be noted that those works only used 7 EEG channels, and we cannot solely rely on accuracy due to the high class imbalance. Also while they used the same dataset as us, there is variability in which subset of PSGs ended up being included in the analysis. Still, these results strongly suggest that PedSleepMAE embeddings contain information relevant to sleep health.

### C. Prader-Willi Syndrome (PWS) Cluster Analysis

We now move beyond 30-second sleep events and demonstrate that PedSleepMAE embeddings can capture more nuanced sleep characteristics that may not be even fully elucidated yet by science. Following the case study in [8], we focus on Prader-Willi syndrome (PWS) using the EHR. For context, PWS is a rare genetic disorder that approximately affects 1 out of 10,000 to 30,000 people, and there has been observations of breathing abnormalities and sleep disorders of PWS patients [14]. Note that compared to earlier labels,

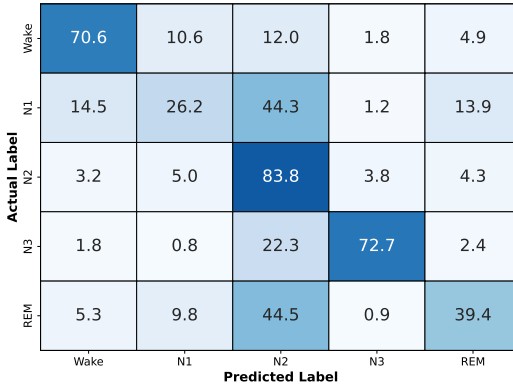

Fig. 3: Sleep scoring is accurate in Wake, N2 and N3, while N1 and REM are often misclassified as N2. Each row adds to 100% in this normalized confusion matrix.

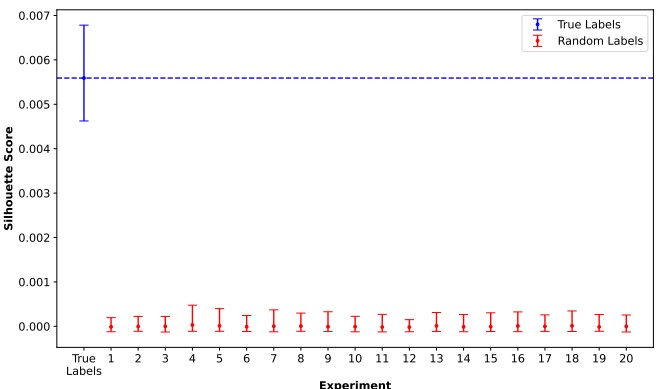

Fig. 4: Comparison of silhouette scores with 95% confidence intervals for PWS/non-PWS and randomly clusters. Higher value is better clustering with +1 being perfect.

detecting sleep characteristics of PWS is much more difficult and nuanced due to many confounding variables and lack of clarity in the exact connection between PWS and sleep.

For this experiment, we decide against training a linear classifier as in Sec. III-B. Automatic detection of apnea, for example, is a meaningful undertaking on its own beyond the fact that it can demonstrate the usefulness of our embeddings. But predicting PWS diagnosis has no clinical use at the end of the day. So instead we perform a cluster analysis on PWS and non-PWS cohorts using silhouette score [15]. Originally used to evaluate clustering algorithms, silhouette score measures how similar a point is to its own cluster compared to other clusters. In general, higher silhouette score indicates better clustering with +1 being perfect, and 0 being that all samples lie on the decision boundary. Silhouette scores are known to suffer from the curse of dimensionality [16], but we sidestep this issue by interpreting scores only relative to each other. We borrow this metric to measure the separation between PWS and non-PWS sleep in PedSleepMAE embedding space.

Cohort 1 has 9,600 PedSleepMAE embeddings from PWS patients, and Cohort 2 has 279,236 embeddings from obese but non-PWS patients. We control for the effect of OSA as in [8]. For computational feasibility, we randomly select 2,500 embeddings from each cohort to compute a silhouette score [17]. This sampling process is repeated 100 times, which is then used to calculate the 95% confidence interval (CI) of the true silhouette score. Same analysis is performed 20 more times with randomly shuffled clusters. The 21 CIs are presented in Fig. 4.

We initially hypothesized that the embeddings would have captured small differences in sleep characteristics that are relevant to PWS. If the features are not related to the PWS labels at all (i.e. under the null hypothesis), the silhouette score calculated from PWS/non-PWS clusters should be indistinguishable from the score calculated from any random two clusters. Therefore, the non-overlapping CIs in Fig. 4 provide evidence supporting our hypothesis. We also conducted a Welch's t-test comparing the true PWS/non-PWS cluster with

each of the 20 random clusters. All 20 t-tests returns t-statistics between 90-93, translating to p-values less than $1 \times 10^{-100}$. While the effect size is small, the statistical significance is robust. Furthermore, since the exact effect of PWS on the sleep signals are not fully understood yet, using generative models to identify potential biomarkers could be an exciting avenue of future research.

## IV. HOW ACCURATE ARE THE GENERATED SIGNALS?

Now that we have established the usefulness of the embeddings, we bring the decoder back into the picture. In this section, we demonstrate how the decoder can be used to retrieve or generate representative examples for a patient, as well as impute missing channels.

### A. Generating and Retrieving Representative Examples

Seminal works in computer vision (e.g. [18]) and natural language processing (e.g. [19]) have shown that with good generative models, arithmetic operations in the embedding space could lead to analogous operations in the original data space. For example, one could take the Euclidean average of face embedding vectors and push it through the decoder to create an "average face image." In comparison, taking the average in the pixel space will not result in a valid face image. Borrowing this idea, we use PedSleepMAE to retrieve and generate representative examples of a patient's sleep.

First, we investigate whether pairwise relationships in the embedding space are maintained in the generated signal space. We check this by measuring pairwise Euclidean distances in both spaces and calculating Pearson's correlation coefficient ($\rho$). $\rho$ ranges from -1 to 1, where 1 indicates a perfect positive linear relationship. Fig. 5 shows a strong positive correlation of 0.93 based on 1,000 samples from a randomly selected patient. While $\rho$ is calculated using all pairwise distances, only 2,000 pairwise distances are plotted to ensure readability. To make sure this was not an outlier patient, we repeated this experiment on 1,000 samples randomly selected from all patients, which yielded $\rho = 0.88$ (Fig. 14 in Appendix D). We

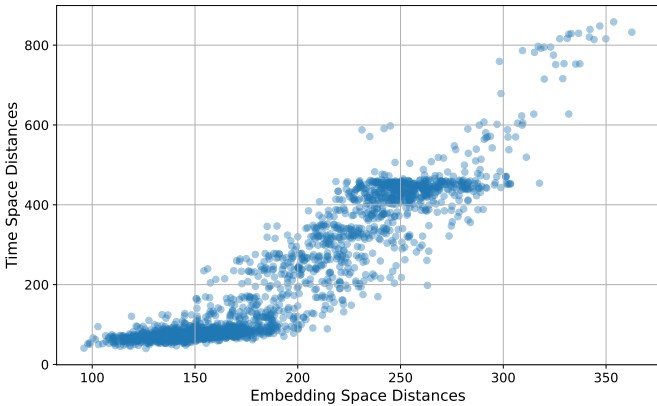

Fig. 5: Pairwise Euclidean distance of the embeddings is highly correlated ($\rho = 0.93$) with pairwise distance of the generated signals. Each point corresponds to a pair of 30s sleep segments. Data from one random patient.

also repeated the process with dynamic time warping (DTW) distance in Appendix D with similar results.

Given this promising quantitative result, we next tackled representative signal generation and retrieval. Let's consider a single sleep stage (REM). We begin by acquiring the embeddings for all REM sleep from a randomly selected PSG, then take the Euclidean average to generate an "average embedding." This vector is then pushed through the decoder to generate an "average 30 seconds of REM sleep" for this PSG (Fig. 6). Albeit noisy, they seem to be valid signals with the right shape, which is not the case if we take the Euclidean average in the original signal space. These synthetic signals can circumvent privacy concerns and be useful toy data in classrooms or be used to develop code quickly in fast-paced projects while waiting for the real medical data to arrive.

Of course, a clinician is unlikely to accept this set of AI-generated signals to make clinical decisions. But now that we have a notion of mean and distance, we can perform a nearest neighbors search to retrieve an actual data sample that is closest to the mean (Fig. 7). On the flip side, we can also find outliers and bring them to the doctor's attention.

Lastly, we showcase a PedSleepMAE-generated "average 30 seconds of sleep with apnea" in Fig. 8. We used a random PSG with at least 100 apnea occurrences to calculate the average embedding. A notable finding is the decreasing SpO2 values and increasing CAPNO levels over time, as well as the fluctuations in C-Flow and RESP abdominal. In patients with apnea, there are frequent interruptions in breathing during sleep, which can lead to periods of reduced blood oxygen levels and increased $CO_2$ levels. During this time, the amount of air flowing from the CPAP machine (if the patient has one) and the effort their body puts into breathing (as measured by elastic bands) may also fluctuate. This event is clearly reflected in Fig. 8. Moreover, the timings are coordinated across channels, which suggest that PedSleepMAE has learned accurate inter-channel information about apnea.

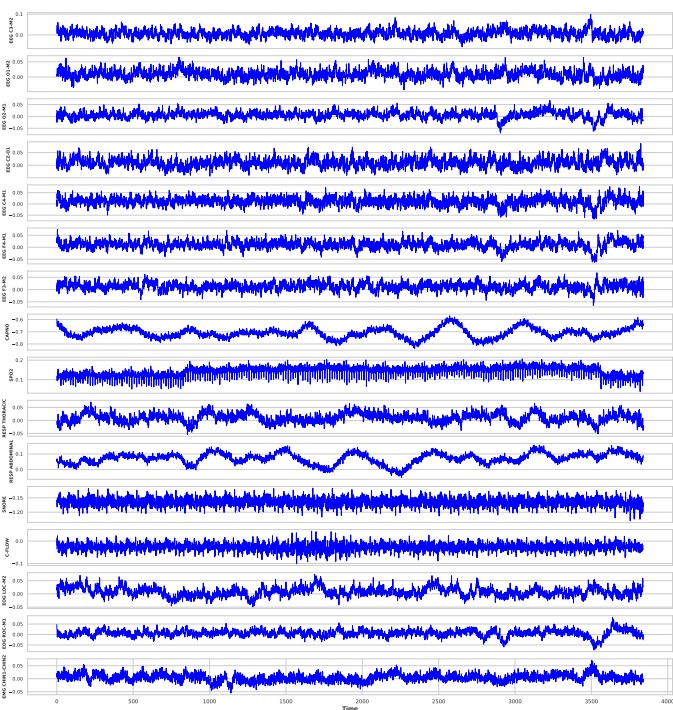

Fig. 6: A 30 seconds of REM sleep generated by PedSleepMAE.

### B. Channel Imputation and Evaluation

Our final experiment is on using PedSleepMAE for missing channel imputation. Recall that during SSL, the model is trained to reconstruct small patches of size 8 ($\approx$ 0.0625 seconds). Generating 30 seconds is a much more difficult task. We systematically remove one channel at a time and let PedSleepMAE reconstruct the missing signal based on the other 15 channels. Table II reports the reconstruction error across 5,000 samples as measured by MSE and DTW. For comparison, a simple mean imputation should theoretically give an MSE of 1 in expectation, because we normalized each channel to 0 mean and 1 standard deviation (SD) in the beginning. It is difficult to make a definite statement due to high SD, but it is promising that the MSE values are much lower than that baseline. We also show an example of the reconstruction for channel EEG F3-M2 in Fig. 9. While there are differences in the amplitude of the signals, the reconstructed signal maintains a similar shape.

### V. CONCLUSIONS

We designed PedSleepMAE, a masked autoencoder (MAE) trained on a large set of pediatric sleep signals. Our transformer-based model learns informative representations of the multimodal data, which includes EEGs, respiratory signals, EOGs and EMG. Our extensive evaluations were both qualitative and quantitative, and we demonstrated a wide range of interesting use cases from apnea detection to information retrieval to data imputation.

While writing this paper, we came across a concurrent work [20] that built a foundation model on an impressively

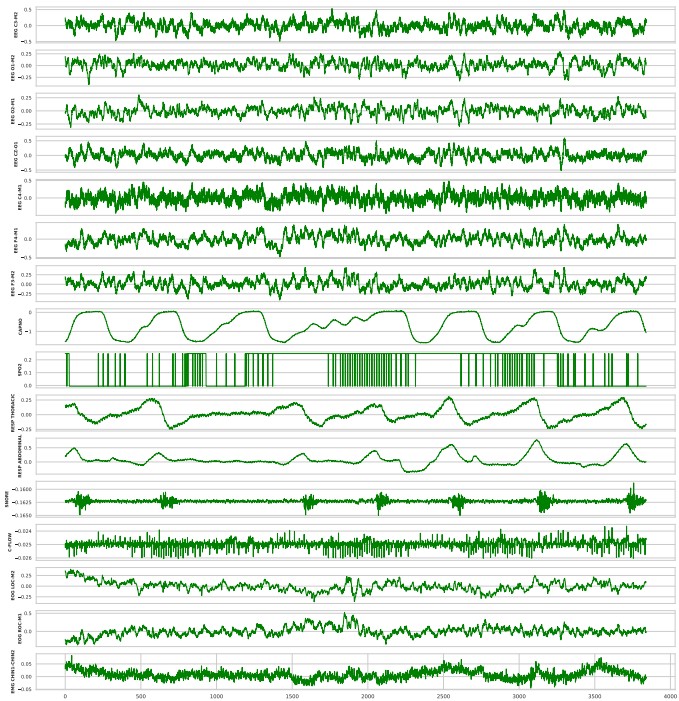

Fig. 7: Real REM sleep example closest to the sample in Fig. 6.

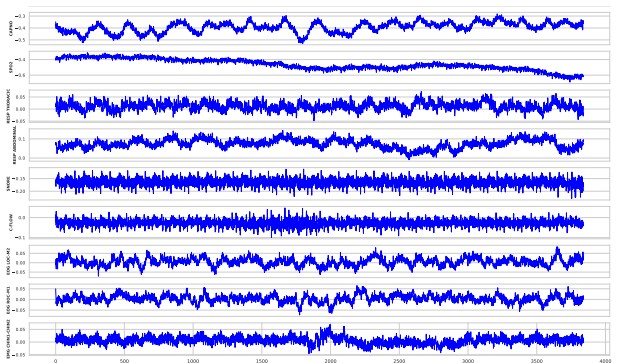

Fig. 8: An average 30 seconds of sleep with apnea generated by PedSleepMAE (EEG channels not shown). Changes in respiratory signals are consistent with apnea.

large private PSG dataset with similar modalities. Our work is differentiated by our use of a public dataset that is focused on pediatric sleep and our design of meaningful and diverse experiments that go beyond sleep scoring and age or gender classification. However, it would be intriguing to try their model on NCH Sleep DataBank, which we defer to future work. Other future directions include trying different generative AI models, especially those that can be trained conditional on patient information derived from the EHR. We also plan to investigate our PWS case study further to come up with biomarkers that can be verified in future clinical experiments.

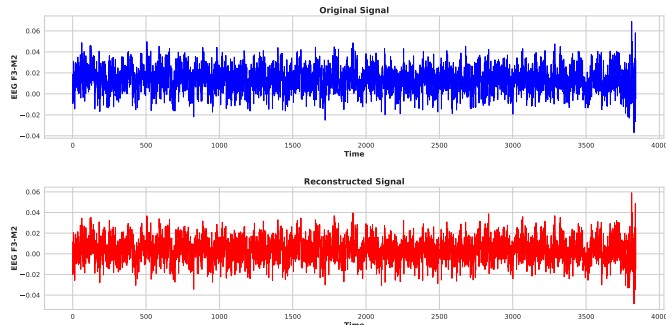

Fig. 9: Original EEG F3-M2 signal (top) and signal imputed by PedSleepMAE (bottom) have very similar waveforms.

TABLE II: Reconstruction error by PedSleepMAE in missing channel imputation. Averaged over 5000 samples.

| Imputed channel | Mean MSE (SD) | Mean DTW (SD) |
| --- | --- | --- |
| EEG C3-M2 | 0.0199 (0.187) | 63.5 (106.4) |
| EEG O1-M2 | 0.0195 (0.137) | 73.9 (94.2) |
| EEG O2-M1 | 0.0156 (0.150) | 42.8 (102.8) |
| EEG CZ-O1 | 0.0241 (0.177) | 65.0 (117.9) |
| EEG C4-M1 | 0.0189 (0.205) | 63.6 (118.2) |
| EEG F4-M1 | 0.00995 (0.0939) | 55.4 (67.9) |
| EEG F3-M2 | 0.0122 (0.0721) | 58.5 (68.8) |
| CAPNO | 0.0581 (0.314) | 114.5 (156.0) |
| SPO2 | 0.0821 (0.648) | 139.9 (229.5) |
| RESP THORACIC | 0.0821 (0.566) | 83.6 (211.6) |
| RESP ABDOMINAL | 0.0760 (0.325) | 116.8 (194.7) |
| SNORE | 0.000876 (0.00644) | 49.5 (23.9) |
| C-FLOW | 0.00178 (0.0125) | 72.9 (25.7) |
| EOG LOC-M2 | 0.0128 (0.0894) | 46.4 (73.6) |
| EOG ROC-M1 | 0.0136 (0.123) | 43.8 (98.9) |
| EMG CHIN1-CHIN2 | 0.0202 (0.152) | 59.5 (127.5) |

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

## APPENDIX A
### MAE PARAMETERS AND SSL STRATEGY

This section discusses the MAE model and pretraining strategy in more detail, following Sec. II-B. The encoder and decoder modules in PedSleepMAE are of the same size. They have three layers of Vision Transformer (ViT) [12] attention blocks, each of which has four attention heads, a multi-layer perceptron (MLP) and normalizing layers. We used the PyTorch implementation of ViT blocks from [21]. The embedding from the encoder is of size (16 channels) × (480 patches) × (64 patch emb dim). Since this is very high-dimensional, we aggregated information from each patch with average pooling in our experiments. This brings down the embedding dimension to $16 \times 480 = 7680$.

TABLE III: Parameters for pretraining MAE.

| Parameter | Value |
| --- | --- |
| Patch size ($p$) | 8 |
| Embedding dimension ($d$) | 64 |
| Masking ratio ($m$) | 50% |
| Optimizer | AdamW |
| Learning rate | 1e-4 |
| Weight decay | 5e-4 |
| Batch size | 64 |
| Total epochs | 600 |
| Iterations per epoch | 2000 |

When pretraining, we consider PSG data in units of 30 seconds of sleep, recorded (or downsampled) at 128Hz across 16 channels. We start by dividing the signal in each channel into patches of $p$ samples. Each patch is then projected into a $d$-dimensional ($d > p$) space. Then, $m\%$ of the patches are masked at random, and the model is trained to reconstruct the masked patches. Loss function is the mean squared error (MSE) between the reconstructed and original signals. Table III summarizes key parameters.

We experimented with different combinations of masking ratios (50%, 75%) and patch sizes (8, 16), and chose the model with the best training and validation loss. These settings result in a total of 76.5 million model parameters.

## APPENDIX B
### ADDITIONAL UMAP VISUALIZATIONS OF EMBEDDINGS

We include additional embedding visualizations in continuation of Sec. III-A. Figs. 10 and 11 are plotted similar to Figs. 2b and 2c in Sec. III-A. Each plot represents one randomly chosen PSG, and each point represents 30 seconds of sleep, colored by different sleep events. In many cases, there are clustering behaviors by sleep events. In Fig. 12, each plot contains random samples from multiple randomly selected PSGs. In this case, we do not observe clusters in the 2-dimensional UMAP space, potentially due to high variability and confounding variables among patients.

## APPENDIX C
### LINEAR CLASSIFIER PARAMETERS

The details in this section support the sleep event classification experiment in Sec. III-B. We split our dataset into 80% training, 10% validation, and 10% testing. For each of our classification tasks, we implemented the classifier as one fully connected feedforward layer in PyTorch [22]. It was then fitted to the training data with batch size 256, AdamW optimizer, learning rate 1e-3, weight decay 1e-5, weighted cross entropy loss and 2000 iterations per epoch for 50 epochs. For binary classification tasks, we chose the threshold for the positive class based on the value that maximized the binary F-1 score on the validation set.

## APPENDIX D
### ADDITIONAL PAIRWISE DISTANCE PLOTS

We provide additional plots and correlation calculations from Sec. IV-A. Figs. 14 and 13 show that regardless of

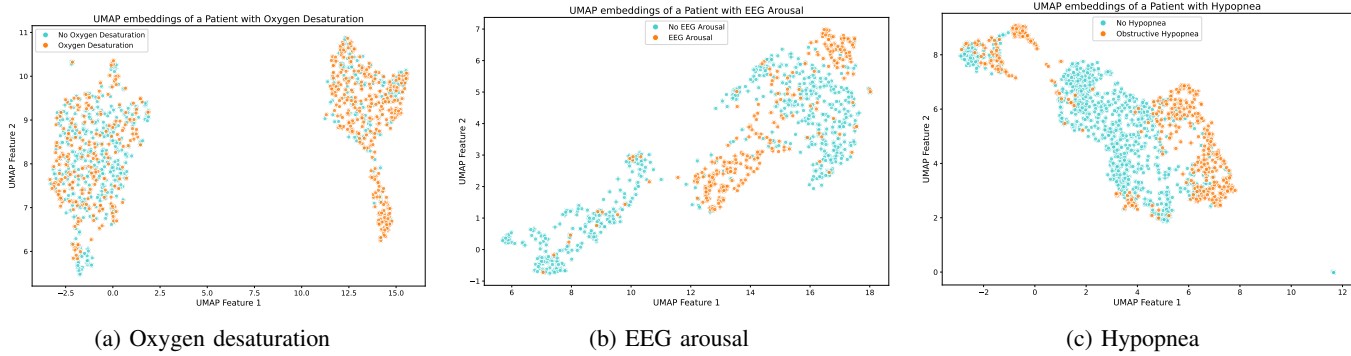

(a) Oxygen desaturation  (b) EEG arousal  (c) Hypopnea

Fig. 10: Each plot represents one PSG. PSGs are randomly chosen from 5 PSGs with most occurrences of the sleep event.

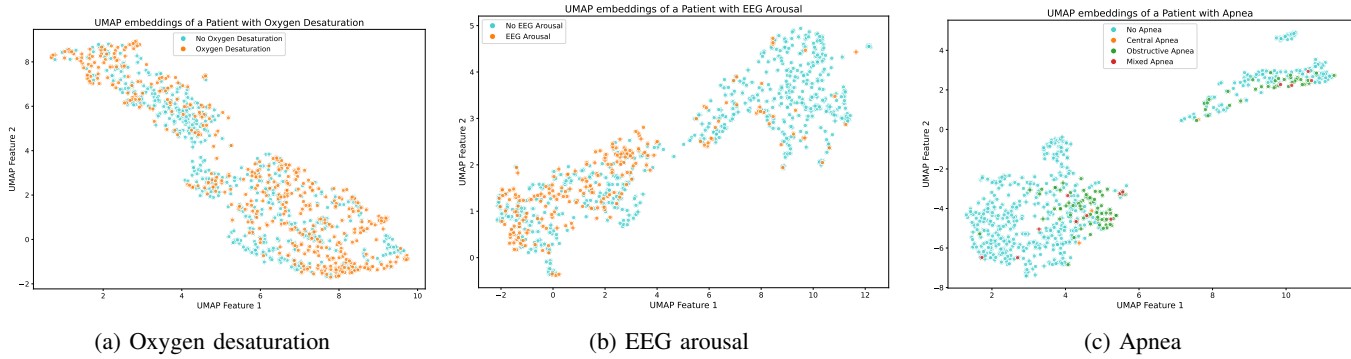

(a) Oxygen desaturation  (b) EEG arousal  (c) Apnea

Fig. 11: Each plot represents one PSG. PSGs are randomly chosen from those with between 5 and 30 cases of the sleep event.

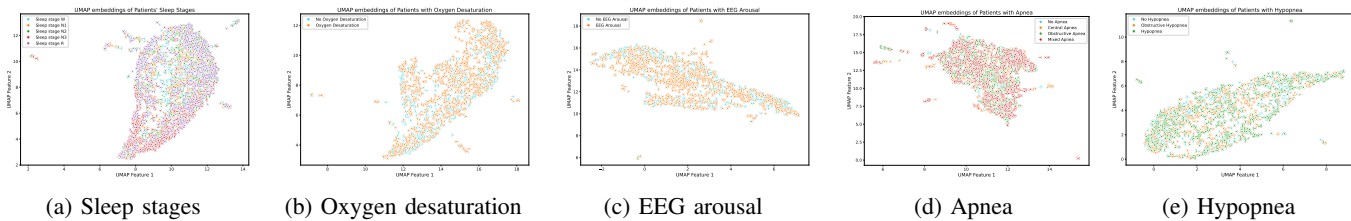

(a) Sleep stages  (b) Oxygen desaturation  (c) EEG arousal  (d) Apnea  (e) Hypopnea

Fig. 12: Unlike previous UMAP visualizations, each plot contains random samples from multiple randomly selected PSGs.

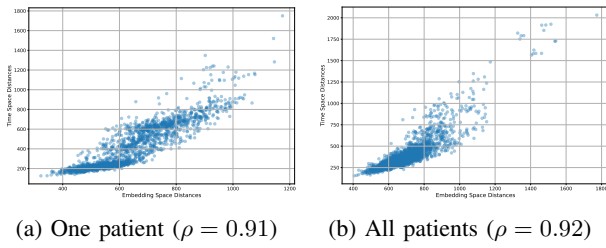

(a) One patient ($\rho = 0.91$)  (b) All patients ($\rho = 0.92$)

Fig. 13: Pairwise distances calculated using DTW.

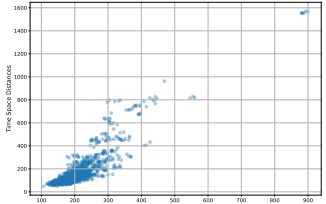

Fig. 14: Pairwise Euclidean distances are highly correlated ($\rho = 0.88$) between embedding space and generated signal space. Data from 1,000 randomly selected samples from multiple random patients.

whether we use Euclidean or dynamic time warping (DTW) distance, or whether we consider a single random patient or random samples from across all patients, pairwise similarities in embedding space are preserved in the generated signal space. We used fast implementation of DTW by [23] and calculated Pearson's correlation coefficient with SciPy [24].

These results provide justification for generating "average"

or representative signals using PedSleepMAE. Additionally, strong linear correlation suggest that a nearest neighbor search done in either the embedding space or the generated signal space would give similar results. Therefore our nearest neighbor search (Fig. 7) is done in the generated signal space.