# OpenReview forum: "PedSleepMAE: Generative Model for Multimodal Pediatric Sleep Signals"
_IEEE.org/EMBS/BHI/2024/Conference — IEEE BHI'24_

### Official Review · Reviewer_52zM · 2024-08-09
**PedSleepMAE: Generative Model for Multimodal Pediatric Sleep Signals**

**Overall Rating:** 7
**Confidence:** 4

**Other Quality Metrics:**

1. Clarity of writing: good. However, the structure of the work makes it somewhat challenging to follow the discourse.
2. Clinical significance: good. However, it would be helpful to see a comparison with other state-of-the-art methods in the results discussion.
3. Methodological novelty: good
4. Experiments and results: good

**Questions For The Authors:**

1. Could you explain how the Masked Autoencoder was adapted for one-dimensional data?
2. How did you handle the input channels? Were the embeddings extracted by processing each channel independently?
3. Have you considered evaluating your method on external and unseen datasets?
4. Could you provide a more detailed explanation of the detection process within your model?
5. Given the imbalanced data, would it be possible to include the Cohen's Kappa metric in your evaluation?
6. Did you apply any post-processing techniques to the generated signals to mitigate noise?
7. Which aspect of the experimental design posed the greatest challenges for you?
8. Is it possible for the proposed tasks (sleep scoring and events detection) to be performed simultaneously? Or do these tasks need to be pre-specified and performed one-by-one?

**Strengths:**

PedSleepMAE is a pioneering generative model specifically designed for pediatric sleep analysis, trained on a large and diverse dataset from Nationwide Children's Hospital, which includes multiple types of sleep signals. This model excels in sleep scoring and detecting events like apnea and oxygen desaturation while also generating realistic sleep signals for tasks such as missing channel imputation and outlier detection. Its embeddings, which are rich in diagnostic information, perform comparably to traditional supervised learning models. PedSleepMAE is notably adept at capturing subtle differences in sleep patterns, including those indicative of rare genetic disorders like Prader-Willi Syndrome. By leveraging a large, publicly available pediatric polysomnography dataset collected in real clinical settings, the model fills a significant gap in health informatics, particularly in pediatric sleep research. Additionally, it is flexible with missing channels and can process various PSG signal types, further enhancing its utility in clinical and research applications.

**Summary Of The Paper:**

The paper introduces PedSleepMAE, a generative model using a masked autoencoder to analyze pediatric sleep data from multichannel EEGs, respiratory signals, EOGs, and EMG. The model performs comparably to supervised learning models in tasks like sleep scoring and apnea detection. Moreover, it captures subtle differences in sleep signals, even identifying characteristics associated with rare genetic disorders and can generate realistic signals for applications such as sleep segment retrieval and missing channel imputation.

**Weaknesses:**

Despite its strengths, PedSleepMAE faces several challenges. The variability and inconsistency in EHR data, stemming from differences in recording and annotation practices across clinical settings, can impact the model's generalizability to other datasets or environments. Additionally, the model's reliance on a specific dataset, the NCH Sleep DataBank, raises concerns about its ability to generalize findings to broader pediatric populations. Privacy and security issues related to the sensitive patient information contained in EHR data are also significant concerns. Furthermore, the high dimensionality of the model's embeddings complicates visualization and interpretation, with techniques like UMAP not always producing clear clusters. The richness of EHR data may introduce confounding variables unrelated to sleep disorders, potentially hindering the model's ability to accurately identify relevant patterns and biomarkers.

---

### Official Review · Reviewer_6LUq · 2024-08-12
**PedSleepMAE: Generative Model for Multimodal Pediatric Sleep Signals**

**Overall Rating:** 6
**Confidence:** 4

**Other Quality Metrics:**

(a) Clarity of writing: Good
(b) Clinical Significance: Fair
(c) Methodological Novelty: Fair
(d) Experiments and Results: Good

**Questions For The Authors:**

In the statement, "Each point represents 30 seconds of sleep, and each plot corresponds to one PSG," what is the definition of 'plot'? It appears that the plot may represent data from all PSGs (since the subsequent description mentions that Fig. 2b shows the top 5 PSGs with the highest occurrences of apnea).

The authors note that while writing this paper, they encountered a concurrent study [20] that developed a foundation model using a notably large private PSG dataset with similar modalities. Including a comparison of the methodologies from this concurrent work in the introduction section would provide valuable context and enhance the study's positioning within the field.

**Strengths:**

The manuscript is well-written and provides comprehensive details on all aspects of analysis and findings. It presents significant work that is highly relevant to the journal. The manuscript presents a comprehensive evaluation of PedSleepMAE, which includes both qualitative and quantitative analyses. The model demonstrates notable accuracy in five-stage sleep scoring and seven binary detection tasks, including oxygen desaturation, EEG arousal, and various subtypes of apnea and hypopnea. Furthermore, the extensive evaluation highlights its versatility in diverse applications such as apnea detection, information retrieval, and data imputation. This thorough assessment underscores the model’s robustness and its potential to address multiple challenges in pediatric sleep signal analysis. TThe figures are well-crafted and effectively illustrate the overall methodology.

**Summary Of The Paper:**

PedSleepMAE is a generative model that effectively integrates multimodal pediatric sleep signals, including EEGs, respiratory signals, EOGs, and EMG, using a masked autoencoder framework. It matches supervised learning models in accuracy for sleep scoring and detecting various sleep abnormalities. The model’s embeddings identify subtle differences associated with rare genetic disorders and it generates realistic signals for applications such as sleep segment retrieval and missing data imputation. This is the first general-purpose generative model for pediatric sleep signals.

**Weaknesses:**

Overall, the manuscript is well-written and presents an important work. The model is well-described but lacks a comparison with some more recent models and methods. A comprehensive comparison would further strengthen the manuscript. The description for Figure 2 should be revised a little to improve user understanding.

---

### Official Review · Reviewer_Ripa · 2024-08-13
**Review: PedSleepMAE: Generative Model for Multimodal Pediatric Sleep Signals**

**Overall Rating:** 6
**Confidence:** 4

**Other Quality Metrics:**

#

- Clarity of writing: ***Good***
- Clinical Significance: ***Fair***
- Methodological Novelty: ***Good***
- Experiments and Results: ***Poor***

**Questions For The Authors:**

- It is difficult to assess the quality of prediction via MSE and DTW given in Table II, without normalizing the signal, provide a percentage error, or provide a comparison against other state-of-art studies, especially when the signals in different channels are not of the same scale. Could the authors provide a MSE metric with normalized signals?

**Strengths:**

- Clean and comprehensive result presentations via multiple visualization methods
- Detailed discussion on hyperparameter selection given in the appendix.

**Summary Of The Paper:**

This manuscript presents an autoencoder based generative model for the following objectives:

- Self-supervised sleep event classification tasks, slightly poorer performance compared to supervised model
- Capture characteristics Prader-Willi Syndrome via subtle differences in sleep signals
- Generate sleep signals for segment retrieval, outlier detection, and data imputation.

**Weaknesses:**

- For sleep event classification tasks, this manuscript present a result with high accuracy and AUROC but ***low F-1 score (6 out of 7 binary tasks have F-1 score lower than 55%)***, indicating the model is barely outperforming random baseline in terms of correctly classifying ‘positive’ cases. In this case of highly imbalanced dataset, good performance in accuracy and AUROC does not align with the objective of classifying positive cases.
- In *Fig. 4*, the silhouette scores from PWS/non-PWS clusters shows a CI around 0.006. This improvement is insignificant in a scale of 0-1 and should not be considered as strong evidence supporting the hypothesis.

    > Therefore, the non-overlapping CIs in Fig. 4 provide strong evidence supporting our hypothesis.
    >

---

### Decision · Program_Chairs · 2024-09-23

Accept